

# The Role of Naphthalene and Its Derivatives in the Formation of Secondary Organic Aerosols in the Yangtze River Delta Region, China

*Fei Ye[1], Jingyi Li[1], Yaqin Gao[2], Hongli Wang[2], Jingyu An[2,3], Cheng Huang[2], Song Guo[4], Keding Lu[4], Kangjia Gong[1], Haowen Zhang[1], Momei Qin[1], Jianlin Hu[1]*

[1] Jiangsu Key Laboratory of Atmospheric Environment Monitoring and Pollution Control, Collaborative Innovation Center of Atmospheric Environment and Equipment Technology, School of Environmental Science and Engineering, Nanjing University of Information Science & Technology, Nanjing, 210044, China

[2] State Environmental Protection Key Laboratory of the Formation and Prevention of Urban Air Pollution Complex, Shanghai Academy of Environmental Sciences, Shanghai 200233, China

[3] Shanghai Key Laboratory of Atmospheric Particle Pollution and Prevention, Department of Environmental Science and Engineering, Fudan University, Shanghai 200438, China

[4] State Key Joint Laboratory of Environmental Simulation and Pollution Control, College of Environmental Sciences and Engineering, Peking University, Beijing, 100871, China

Correspondence to: Jingyi Li (jingyili@nuist.edu.cn), Jianlin Hu (jianlinhu@nuist.edu.cn)



**Abstract.** Naphthalene (Nap) and its derivatives, including 1-methylnaphthalene (1-MN) and 2-
methylnaphthalene (2-MN), serve as prominent intermediate volatile organic compounds (IVOCs)
contributing to the formation of secondary organic carbon (SOC). In this study, the Community
Multi-Scale Air Quality (CMAQ) model coupled with detailed emissions and reactions of these
compounds was utilized to examine their roles in the formation of SOC and other secondary
pollutants in the Yangtze River Delta (YRD) region during summer. Remarkably, significant
underestimations of Nap and MN concentrations (by 79% and 85%) were observed at the Taizhou
site. To better capture the temporal variations of Nap and MN, their emissions in the YRD region
were scaled up by a factor of 5 and 7, respectively, with constraints based on field measurements.
After adjusting their emissions, Nap concentrations reached 27 ppt in the YRD, accounting for 4.1%
and 9.1% (up to 13.7%) of total aromatics emissions and aromatic-derived SOC, respectively. 1-
MN and 2-MN were relatively low, with an average of 3 and 6 ppt in the YRD, and contributed
3.1% of aromatic-derived SOC. The influences of Nap and MN oxidation on ozone and radicals
might be trivial on a regional scale but were not negligible when considering daily fluctuations,
particularly in Shanghai and Suzhou. This study emphasizes the high SOC formation potentials of
Nap and MN, which may pose environmental risks and adverse health.
**1 Introduction**

Secondary organic aerosols (SOA) are formed from the condensation and multiphase

evolution of less volatile organic compounds (VOCs), which can be directly emitted or produced
from the oxidation of higher volatile organics in the atmosphere. SOA not only affects visibility
and human health but also has impacts on the climate directly by absorbing and reflecting solar
radiation and indirectly by affecting cloud formation (Chen et al., 2016; Zhang and Ying, 2012).
Semi-volatile and intermediate-volatile organic compounds (S/IVOCs) have been identified as the



key precursors of SOA (Robinson et al., 2007; Hu et al., 2022). IVOCs are categorized by small
polycyclic aromatic hydrocarbons (PAHs), intermediate-length alkanes (e.g. n-hexadecane), and
phenols(Pye and Seinfeld, 2010). PAHs are organic compounds containing multiple aromatic rings.
China was responsible for the highest annual PAH emissions at 114 Gg with a portion of 22% of
global total PAH emissions in 2004(Zhang and Tao, 2009). Naphthalene (Nap) and
methylnaphthalene (MN), such as 1-methylnaphthalene (1-MN) and 2-methylnaphthalene (2-MN),
are the most abundant airborne PAHs (Chen et al., 2016; Fang et al., 2021), which are mainly
emitted from the combustion of fossil fuels, biomass burning, and industrial sectors (Fang et al.,

2021).

Chamber studies have identified the gas- and particle-phase products from Nap reacting with
hydroxyl radical (OH·) (Huang et al., 2019). Ring-retaining products (e.g., 1,4-naphthoquinone)
with lower volatilities are dominant under low nitrogen oxide ($NO_x$) conditions, and ring-opening
products (e.g., 2-formylcinnamaldehyde) with higher volatilities are dominant in the presence of
high $NO_x$. Chan et al. (2009) evaluated the SOA yields of Nap, 1-MN, 2-MN, and 1,2-dimethyl
naphthalene in chambers and applied these yields to estimate SOA formation from primary
emissions of diesel engines and wood burning. The SOA yields were 55–75% under low-$NO_x$
conditions at a total organic aerosol loading of 15 µg m$^{-3}$, which was more efficient than high-$NO_x$
conditions (25–45%). In the photo-oxidation period of less than 12 h, these PAHs produced 3–5
times more SOA than light aromatic compounds and were responsible for up to 54% of total SOA
from the oxidation of diesel emissions. Huang et al. (2019) applied a tracer method and discovered
that 14.9% of SOA was owing to the oxidation of Nap and MN in the afternoon during the
wintertime haze in Beijing. Shakya and Griffin (2010) also reported 36–162 kg day$^{-1}$ of SOA
production from the mobile source emitted PAHs (including Nap, 1-MN, and 2-MN) in Houston



based on the yields from their study and that of Chan et al. (2009). Based on the yield from Shakya
and Griffin (2010), Liu et al. (2015) showed that Nap contributed 8–52% of the total SOA
originating from benzene, toluene, C2-benzene, C3-benzene, C4-benzene, and Nap in light-duty
gasoline vehicle exhausts. All these experimental findings demonstrate the significant role of Nap
and MN in SOA formation in the environment with anthropogenic influences dominated. However,
these results might not accurately reflect the actual atmospheric conditions due to the simplicity of
reaction conditions and the limited precursors involved in chamber studies (Ling et al., 2022).
Numerical models have been developed and utilized to assess the contribution of S/IVOCs to
SOA (Hayes et al., 2015; Pye and Seinfeld, 2010; An et al., 2023). Zhang and Ying (2011) showed
that PAHs emitted from anthropogenic sources could produce SOA mass as much as 10% of that
from the traditional light aromatics or around 4% of total anthropogenic SOA by using the
Community Multiscale Air Quality (CMAQ) model. However, the products from several explicit
PAH species (Nap, MN, dimethyl naphthalene, ethyl naphthalene, acenaphthylene, acenaphthene,
fluorene, phenanthrene, fluoranthene) were lumped rather than separated for their contributions to
SOA due to limited experimental data. Pye and Pouliot (2012) utilized the CMAQ model and
tracked 10% of peroxy radicals produced from the ARO2 (lumped aromatics in CMAQ) and OH·
reaction as for that of Nap without considering the emissions and the accurate OH· reactivity of
Nap. According to Cohan et al. (2013), the modeled SOA increased by roughly 1–10% when Nap
emissions from on-road gasoline and diesel vehicles were considered. Their simulations showed a
lower bound in the SOA production from Nap due to underestimations in the emission inventory
in the South Coast Air Basin of California. Majdi et al. (2019) found that Nap and MN contributed
2.4% to the total organic aerosol (OA) originating from wildfires over the Euro-Mediterranean



region during the summer of 2007 by using a 3D chemistry transport model (CTM). The
contributions of Nap and MN to SOA over a regional scale in China had not been quantified.

In this study, SOA formation from Nap, 1-MN, and 2-MN in the Yangtze River Delta (YRD)

region during the EXPLORE-YRD (EXPeriment on the eLucidation of the atmospheric Oxidation
capacity and aerosol foRmation, and their Effects in Yangtze River Delta) campaign period (May
20 − June 18, 2018) was investigated with an updated CMAQ model. Emission inventories of Nap,
1-MN, and 2-MN were estimated based on different sources and methods and validated against
observations. After that, the influences of Nap and MN on secondary organic carbon (SOC), ozone
($O_3$), and radical concentrations in the locations with high concentrations and at the regional scale
were examined separately. The newly added SOA parameterizations for 1-MN and 2-MN were
fitted by both two-product and one-product methods to compare the differences. We find that Nap
and its derivatives, although accounting for a small fraction of emitted aromatics (5.1%),
contributed 12.1% of aromatic-derived SOC in the YRD.
**2 Methods**
**2.1 Modified SOA formation pathways of MN**

The CMAQ model version 5.2, coupled with the SAPRC07tic atmospheric chemical

mechanism and the AERO6i aerosol module, was updated to include the oxidation of 1-MN and
2-MN by OH· and the corresponding SOA formation pathways. In the original CMAQ model, Nap
reacts with OH· to form SOA under low- and high-$NO_x$ conditions, which are represented by two
different counter species PAHHRXN and PAHNRXN, respectively (Fig. S1). Similar to Nap, 1-
MN, and 2-MN were treated explicitly as reacting with OH· and forming SOA counter species
under high $NO_x$ (aMPAHNRXN and bMPAHNRXN) and low $NO_x$ (aMPAHHRXN and
bMPAHHRXN), along with other products following Zhang and Ying (2012). These counter



species were used to calculate the production of SOA through gas-particle partitioning based on
yields ($\alpha_i$) and partitioning coefficients ($K_{\mathrm{om},i}$, m$^3$ µg$^{-1}$) of condensable organic products derived
from chamber experiment data. The detailed descriptions of gas-particle partitioning to fit SOA
yield through one-product and two-product methods are depicted in the Supplement.

In gas-particle partitioning of the original CMAQ model, a two-product method (SV_PAH1

and SV_PAH2) was used to represent the SOA formation from Nap under high-NO$_x$ conditions,
which are denoted as APAH1J and APAH2J respectively (Fig. S1a). Under low-NO$_x$ conditions,
a one-product method was used to represent the SOA formation from Nap, denoted as APAH3J.
It was assumed that APAH3J (a yield of $\alpha_3$) was non-volatile and resided in the particle phase.
Similar to Nap, a two-product method for the oxidation products of 1-MN was added under high-
NO$_x$ conditions as shown in Fig. S1b, with the SOA species denoted as AaMPAH1J and
AaMPAH2J. A one-product method to characterize the oxidation products of 1-MN was also
applied to compare the difference caused by the fitting approach. As shown in Fig. S1c, the semi-
volatile organic product SV_aMPAH1' undergoes equilibrium partitioning to form SOA
(AaMPAH1J'). Under low-NO$_x$ conditions, a non-volatile SOA product AaMPAH3J is formed by
the oxidation of 1-MN. The SOA pathways of 2-MN follow 1-MN, with the corresponding SOA
products of AbMPAH1J, AbMPAH2J, and AbMPAH3J, respectively. In addition, all semi-
volatile SOA products undergo condensed-phase oligomerization reactions at the same rate of
APAH1J and APAH2J and produce non-volatile oligomers (AOLGAJ) that belong to the
anthropogenic source. Other processes and parameters involved in the newly added SOA pathways
for 1-MN and 2-MN, such as the dry and wet deposition and the molecular weight of the oxidation
products were set to be the same as Nap due to limited experimental data. Details of all the
parameters, i.e., $\alpha_i$, $K_{\mathrm{om},i}$, and $\Delta H_{\mathrm{vap},i}$ are summarized in Table S1.





**2.2 Model application**

The simulation domain, which covers Jiangsu, Zhejiang, Anhui, Shanghai, and neighboring

provinces, has a horizontal resolution of 4 km × 4 km (238 × 268 grids) and a vertical structure of

18 layers as shown in Fig. S2. Details of the domain setup can be found in previous studies (Li et

al., 2021; Li et al., 2022). The meteorological field was predicted by the Weather Research and

Forecasting (WRF) model version 4.0 with the ECMWF Reanalysis v5.0 (ERA5) reanalysis data

as the inputs. More details about the WRF configuration were summarized by Wang et al. (2021).

A spin-up of two days was used to minimize the influence of initial conditions.

Biogenic emissions were generated from the Model for Emissions of Gases and Aerosols

from Nature (MEGAN) version 2.1 (Guenther et al., 2012). Open biomass burning emissions were

based on the Fire INventory from the National Center for Atmospheric Research (FINN)

(Wiedinmyer et al., 2011). Anthropogenic emissions were generated from the updated 2017

emission inventory for the YRD (Cheng et al., 2021) and the Multi-resolution Emission Inventory

for China (MEIC, http://www.meicmodel.org, last access: 1 June 2023) for the rest of the domain.

Currently, there is no available data to use in more localized sources in China. The detailed

emissions of 1-MN and 2-MN of different sources were calculated from the US EPA

(Environmental Protection Agency) repository of organic gas and PM speciation profiles of air

pollution sources (SPECIATEv5.2) and information reported by An et al. (2021) and Li et al.

(2014). See the Supplement for more details about the calculating process. There were two sets of

emission data consisting of different Nap and MN emissions in the YRD. The emis-orig used the

original Nap emissions from the 2017 YRD inventory and the calculated MN emissions. We show

later that Nap and MN were underestimated in emis-orig and required an adjustment in their

emissions to capture the observed concentrations. Therefore, the anthropogenic emissions of Nap



155 and MN in the YRD region from emis-orig were multiplied by 5 and 7, respectively, and

156 unchanged in other regions in the emis-adjust case. All the emission ratios applied in this study

157 are shown in Table S2. According to Fig. S3, Nap and MN emissions were mainly located in

158 Shanghai, southern Jiangsu, and parts of Zhejiang in the YRD region. After adjustments, the total

159 Nap and MN emission rate over the YRD region in emis-adjust (3.9 kg day$^{-1}$) was approximately

160 fourfold higher than that in emis-orig (0.9 kg day$^{-1}$). The total MN emission rate over the YRD

161 region in emis-adjust was 0.9 kg day$^{-1}$ and was lower than that of Nap. For emis-adjust, the

162 dominant source of MN was residential-related (47.0%), followed by industry process (25.8%)

163 and on-road transport (20.8%). On-road transport contributed the most to Nap emissions in both

164 emis-orig (78.2%) and emis-adjust (87.5%). It should be noted that the source contributions of

165 Nap and MN may be influenced by the uncertainties in the source profiles.

166  Table S3 lists the scenarios conducted in this study. In case-1product-orig, the anthropogenic

167 emissions in the YRD used emis-orig with default Nap and added MN emissions, and the SOA

168 parameterization for MN was fitted by the one-product method. To assess the impacts of different

169 SOA parameterizations, the case-2products-orig shared the same setting with case-1product-orig

170 except that a two-product method for MN-generated SOA was employed. Both case-1product and

171 case-2products used emis-adjust as the emission inventory but different SOA parameterizations

172 for MN. In all, the contributions of Nap, 1-MN, and 2-MN to the aromatic SOC were estimated

173 based on different emission inventories and two SOA parameterization schemes. To evaluate the

174 effects of Nap, 1-MN, and 2-MN on O$_3$, SOC, and radical concentrations, their emissions in case-

175 1product were set to zero and named base1.

176 **2.3 Observation data for model validation**



In May-June 2018, the EXPLORE-YRD field campaign was launched at a rural site in
Taizhou (32.558°N, 119.994°E) and simultaneously monitored VOCs (including Nap and MN),
$O_3$, $NO_x$, SOC, OH·, hydroperoxy radical ($HO_2\cdot$), and other various pollutants, which provides a
good opportunity for model validation and understanding the evolution of air pollution in the YRD
(Wang et al., 2020; Huang et al., 2020; Yu et al., 2021; Gao et al., 2022). Details of the
measurement method and accuracy for each species refer to these references. The simulated
MDA8 $O_3$, fine particulate matter ($PM_{2.5}$), sulfur dioxide ($SO_2$), nitrogen dioxide ($NO_2$), and
carbon monoxide (CO) were also compared with the observations from the National Real-Time
Urban Air Quality Release Platform of the China Environmental Monitoring Center
(http://106.37.208.233:20035/, last access on May 17, 2023) in Suzhou, Nanjing, Hangzhou, Hefei,
and Shanghai cities as shown in Fig. S2. The statistical metrics including NMB, NME, and r were
calculated for several air pollution species. The model performance benchmarks followed the
recommendations by Emery et al. (2017) and are listed in Table S4. The meteorological parameters
predicted by WRF have been examined to be robust during the same episode by Wang et al. (2021).
**3 Results**
**3.1 Model validation**
Fig. 1 and Fig. S4 show the comparison of observed and simulated hourly variations of Nap,
MN, $O_3$, organic carbon (OC), and $PM_{2.5}$ at the Taizhou site during the study period. As shown in
Fig. 1, in the original settings, the concentrations of Nap were largely underestimated in emis-orig
by 79% compared with the observations, with the value of NMB being -0.79. In contrast, emis-
adjust better represented the temporal variations of Nap (NMB=0.01, r=0.68) than emis-orig, with
the averaged concentration increased by 375% and more comparable to the observations. The
concentrations of MN simulated by emis-adjust (1.40E-2 ppb) were also comparable to the





observations (1.50E-2 ppb) and showed a good correlation with the observations (r=0.59). For
other species, the concentrations of OC and PM$_{2.5}$ were also improved in emis-adjust compared to
that of emis-orig, although they were underestimated in both scenarios. The NMB and NME of
PM$_{2.5}$ satisfied the benchmark recommended by Emery et al. (2017), while the NMB of the
maximum daily 8-hour average (MDA8) O$_3$ exceeded the benchmark. Table S5 shows that the
concentrations of NO$_2$ and nitric oxide (NO) were underestimated at the Taizhou site suggested by
the negative NMB values. The simulated OH radicals compared well with the observation while
the concentrations of HO$_2$· were underestimated at the Taizhou site (Fig. S5). The predicted
concentrations of MDA8 O$_3$, PM$_{2.5}$, SO$_2$, NO$_2$, and CO were examined as shown in Table S4.
Overall, the model agreed well with observations in most of the cities except for a significant
underestimation of MAD8 O$_3$ in Shanghai. We chose the results from case-1product and case-
2products using emis-adjust as the emission data in the subsequent analysis.
**3.2 Influences of Nap and MN on SOC in Taizhou**

Figure 2 depicts the diurnal variations of emissions and concentrations of Nap, 1-MN, and 2-

MN, as well as the corresponding SOC products SOC-Nap, SOC-1MN, and SOC-2MN at the
Taizhou site in both case-1product and case-2products. The emissions of Nap,1-MN, and 2-MN
exhibited a bimodal pattern. For Nap, the bimodal characteristics were the most pronounced,
accompanied by two peaks that occurred between 8:00~9:00 and 16:00~17:00, respectively. This
was likely attributed to the dominant source of Nap from transport as described in Sect. 2.2. Nap
and MN concentrations were relatively low during the daytime and peaked in the morning and at
night, which was caused by the fast photochemical removal and increased dilution during the
daytime, along with the facilitated accumulation due to low mixing heights at night (Cohan et al.,
2013; Huang et al., 2019). The concentrations of SOC generated by Nap, 1-MN, and 2-MN were





high during the daytime, especially from 10:00 to 15:00. This was attributed to the removal of Nap
and MN by OH radicals to form SOC. The potential removal by nighttime nitrate radicals ($NO_3$)
was negligible in this study, leading to a certain degree of declining trend for SOC formation at
night. Nap-derived SOC was the most abundant, followed by SOC from 2-MN (SOC-2MN) and
1-MN (SOC-1MN). This is attributed to the combined effects of the OH· reactivity, SOA yields,
as well as abundances of the three compounds (Li et al., 2017; Yu et al., 2021). Apart from the
highest emissions of Nap, Nap is also more reactive with OH· and has the highest SOA yield in
case-2products compared to the other two species. In case-1product, although the SOA yields of
MN are the highest, the OH· reaction rate with Nap is faster than MN. The SOC generated by MN
in case-2products was lower than that in case-1product due to the lower SOA yield of MN applied
in case-2products as shown in Table S1.

Figure 3 shows the contributions of major aromatic species, i.e., Nap, 1-MN, 2-MN, 1,2,4-

trimethyl benzene (B124), xylene (MPO), benzene (BENZ), toluene (TOLU), aromatics with $k_{OH}$
(reaction rate constant with OH·) $< 2 \times 10^4$ ppm$^{-1}$ min$^{-1}$ (ARO1) and ARO2MN' (ARO2 excluding
Nap and MN) to the total emissions of aromatics and the aromatic-derived SOC in both case-
1product and case-2products at the Taizhou site. Among all the species, ARO2MN', MPO, and
B124 showed the largest fraction of emissions, accounting for 58.6%, followed by ARO1 and
TOLU (31.8%), and BENZ (6.3%). Nap and MN contributed the least to the total aromatic
emissions, with Nap to be the most abundant species. The daily average SOC produced from all
the aromatics was quite similar in case-1product and case-2products, which were 102.0 and 100.7
ng m$^{-3}$, respectively (Fig. S6). The contribution of ARO2MN', MPO, and B124 to the total
aromatic-derived SOC was the most significant, which was 45.2–45.8%. Nap showed a remarkable
contribution to SOC, accounting for 8.7–8.8%, although it only made up 2.6% of the total emitted





aromatics. 2-MN was also an important SOC precursor, contributing to 1.3–2.2% of the aromatic-
derived SOC. 1-MN was the least emitted aromatic compound, accounting for 0.2% of the total
aromatic emissions and less than 1.0% of the aromatic-derived SOC. All of Nap, 1-MN, and 2-
MN had the same trait of contributing much more to SOC than to SOC precursor emissions,
especially for Nap. The total contributions of MN and Nap to SOC were higher than that of BENZ,
even though their emissions were significantly lower than BENZ. Similar results were also found
in field campaigns conducted in Guangzhou (Fang et al., 2021) and Beijing (Huang et al., 2019)
where Nap and MN showed higher contributions. Compared to BENZ and other single-ring
aromatics, Nap and MN belong to IVOCs with lower saturation vapor pressure, which is more
likely to generate SOA through coagulation and absorption (Gao et al., 2021; Zhao et al., 2014).
Thus, their considerably higher SOA yields and reactivity with OH· lead to an important
contribution to SOA formation. In general, we found that 3.3% of aromatic emissions from Nap
and derivatives could contribute up to 11.7% SOC generated from aromatics at the Taizhou site.
**3.3 Regional distributions of Nap and MN and the influences on secondary pollutants**

In the YRD, the average contribution of Nap to aromatic emissions was 4.1% (Fig. S7), while

the Nap-derived SOC accounted for 9.0% and 9.1% of the total SOC generated by aromatics in
case-1product and case-2products, respectively. We found extremely high contributions of Nap-
derived SOC in areas with high Nap emissions (Fig. S8), reaching up to 13.7% in case-2products.
2-MN constituted 0.6% of the total aromatic emissions and contributed up to 3.8% of the aromatic-
derived SOC in case-1product. Among the three PAHs, 1-MN showed the lowest emissions (about
0.4% of the aromatic emissions) and the smallest regional average contribution to SOC (0.6–0.9%).
The SOC derived from MN in case-2products was approximately 38% lower than that in case-
1product across the entire YRD region (Fig. S8), while $O_3$ and the total SOC showed minor





differences in the two cases with different SOA parameterization of MN (Fig. S9). In general, the
concentrations of SOC produced by the three PAHs in case-1product were higher than those in
case-2products, which may minimize the discrepancy between the simulated and observed OC
given the existing underestimation of OC at least in Taizhou, as shown in Fig. 1 and Fig. S6.
Therefore, we opted for the results from case-1product in the subsequent analysis.
The accurate reproduction and quantitative constraints of Nap and MN are crucial for
understanding the atmospheric oxidation capacity in model simulations. The relative differences
between base1 and case-1product were calculated to evaluate the effects of Nap, 1-MN, and 2-MN
on $O_3$, SOC, and radical concentrations. As shown in Fig. 4a, the SOC concentrations over the
YRD region increased by approximately 1.0% on average, with the most significant increase
observed in areas with high emissions of Nap and MN, such as Shanghai and southern Jiangsu
Province, reaching up to 1.7%. The impact on $O_3$ was relatively limited, with a maximum increase
of 0.3%. Similar to SOC, the spatial distribution of $O_3$ variations was consistent with that of Nap
and MN emissions. When Nap and MN oxidation was considered in the model, $HO_2\cdot$ concentration
was enhanced across the domain by up to 1.6% (in Shanghai), due to the production of $HO_2\cdot$
through the reaction of Nap and MN with $OH\cdot$. However, the variations in $OH\cdot$ concentration
exhibited regional heterogeneity, with a maximum increase of 0.8% (in Shanghai) and a maximum
decrease of 0.3% (in Wenzhou). The areas with elevated $OH\cdot$ coincided with the locations
experiencing notable increases in $O_3$. As an $OH\cdot$ source in the troposphere, the photolysis of $O_3$
produces electronically excited $O(^1D)$ atoms that react with water molecules to form fresh $OH\cdot$
(Tan et al., 2019; Qin et al., 2022). Moreover, the areas with elevated $OH\cdot$ also exhibited a
significant increase in $HO_2\cdot$. $HO_2\cdot$ can react with $O_3$ to produce $OH\cdot$, thereby offsetting the $OH\cdot$
consumption by Nap and MN oxidations (Zhu et al., 2020). In the areas with decreased $OH\cdot$, the



increase of $O_3$ and $HO_2\cdot$ was not significant, resulting in fewer newly generated $OH\cdot$ to compensate
for the $OH\cdot$ consumption by Nap and MN.
To minimize the potential obfuscation of the true magnitude by the episode-average variation,
the hourly relative differences of SOC, $O_3$, and radicals at the Shanghai and Suzhou sites, which
exhibit significant variations, are depicted in Fig. 4b and Fig. 4c, respectively. Overall, the
influences of Nap and MN varied daily. At the Shanghai site, the most pronounced effects of $OH\cdot$
and $HO_2\cdot$ were observed, with increases of up to 1.7% and 3.7%, respectively. At the Suzhou site,
the maximum daily variations of $OH\cdot$ and $HO_2\cdot$ (1.5% and 2.9%) were marginally lower than those
in Shanghai; whereas, the maximum daily variations of SOC and $O_3$ were elevated by 3.0% and
1.1% at the Suzhou site, respectively. Consequently, the influences of Nap and MN on SOC, $O_3$,
and the atmospheric oxidation capacity were substantial at the daily scale in those regions.
**4 Discussion**
Our results revealed that the contributions of Nap and MN to the total aromatic emissions
were minimal, which were 5.1% in the YRD and 3.3% at the Taizhou site. However, the SOC
produced by Nap and MN constituted 12.1% of the total aromatic-derived SOC in this region and
11.7% at the Taizhou site. Given the overestimation of other aromatic species in the current model
(Table S5), the contributions of Nap and MN to aromatic SOC might be underestimated. Yu et al.
(2021) demonstrated an augmented fraction of SOC derived from a yield method to that using the
EC tracer method after the inclusion of Nap and MN oxidation (from 25.3% to 39.5%) during the
same episode at the Taizhou site. That is to say, Nap and MN contributed 35.9% of the total SOC
estimated by using the SOA yield multiplied by the consumption of VOCs, which was higher than
the value (11.7%) in this study. Other field studies also found significant SOA formation from Nap
and MN among aromatics in the Pearl River Delta region (12.4%) (Fang et al., 2021) and in Beijing





during haze days (10.2±1.3%) (Huang et al., 2019), with relatively smaller contributions to
emissions by 2% and 7%, respectively. This study highlights the crucial roles of Nap and MN,
which exhibit high SOA production potentials with trace amounts emitted into the atmosphere. In
addition, the average concentrations of Nap and MN in this study were 27 and 9 ppt during summer
over the YRD region (Fig. S8), respectively. Previous studies have confirmed that the
concentrations of Nap and MN exhibited a seasonal variation, with maxima in winter and minima
in summer, attributed to the increased heating and cooking activities in households during the cold
season (Tang et al., 2020; Huang et al., 2019; Fang et al., 2021). Consequently, the ambient
concentration of Nap and MN, along with the potential SOA production may be more severe in
winter. Cleaner fuel types and household cleaning products are recommended for vehicular and
domestic usage.

The urgent demand for enhancing the simulation and assessment of Nap and MN chemistry

is necessitated. Firstly, the characterization of Nap and MN from local sources and additional field
observations are indispensable to reduce the disparities between the modeled and observed Nap
and MN concentrations. Secondly, the SOA parameterizations of Nap and MN, including the
enthalpy of vaporization and SOA yields, are derived from limited chamber experiments and
require further validation. Previous studies have reported that the SOA yields obtained from
chamber studies were contingent on OH· exposure, $NO_x$ levels, relative humidity, and seed
particles, which may not represent the actual atmospheric conditions (Yu et al., 2021; Ling et al.,
2022). Thirdly, chlorine radicals (Cl), $NO_3$ radicals, and $O_3$ also play an important role in the
atmospheric reactions of Nap and MN (Cohan et al., 2013; Matthieu et al., 2014; Riva et al., 2015;
Wang et al., 2005; Aleman, 2006), which were missing in the current study due to the lack of
parameterization.  The formation of gas- and particle-phase products through reactions between




Cl atoms and Nap has been confirmed. For instance, chloronaphthalene and chloroacenaphthenone
have been identified as potential SOA markers for the Cl-initiated oxidation of Nap in the ambient
atmosphere (Riva et al., 2015). As important sources of Cl atoms, abundant nitryl chloride ($ClNO_2$)
and molecular chlorine ($Cl_2$) were attributed to sea salt, coal combustion, biomass burning (Le
Breton et al., 2018), and urban-originated transports (Li et al., 2021; Tham et al., 2013).
Consequently, the Cl-initiated SOA formation process may be pronounced in specific regions,
such as the marine boundary layer and industrial areas. Using the rate constant of Cl with Nap
(($4.22\pm0.46$)$\times10^{-12}$) (Matthieu et al., 2014) and corresponding SOA yields ($0.91\pm0.05$) (Riva et
al., 2015), which is approximately three times higher than those determined from OH-initiated
oxidation (Chan et al., 2009; Shakya and Griffin, 2010), we estimated the potential SOA formation
from the reaction of Nap and Cl atoms via a yield method (Huang et al., 2019; Yu et al., 2021).
Assuming a global average Cl concentration of $1\times10^4$ molecules $cm^{-3}$ and a tropospheric lifetime
of 275 days as determined by Matthieu et al. (2014), SOA generated from Nap initiated by Cl
atoms is three times higher than that from the oxidation by OH· with a 12-h average daytime
concentration of $2\times10^6$ molecules $cm^{-3}$ and a tropospheric lifetime of 6 hours. This suggests that
the omission of Cl-initiated chemistry in this study might lead to an underestimation of Nap-
derived SOA by approximately 75%. Given the underestimation of anthropogenic chlorine
emissions in China (Li et al., 2021; Choi et al., 2020), further studies are recommended to estimate
chlorine emissions with finer spatial resolution and the impacts on Nap SOA under
atmospherically realistic conditions. Lastly, a precise depiction of Nap and MN chemistry is
crucial for gaining a deeper understanding of the health implications of these noxious compounds.
The health risks associated with inhalation exposure to outdoor Nap and other PAHs have been
assessed by calculating the incremental lifetime cancer risk (ILCR) values in China and the United



States (Han et al., 2020; Zhang et al., 2016). Nonetheless, there has been no systematic evaluation
of the health risks resulting from exposure to PAH-derived SOA and by-products, despite previous
studies verifying the toxicological impacts (e.g. oxidation potential, OP) of Nap-derived SOA
(Lima de Albuquerque et al., 2021; Wang et al., 2018; Tuet et al., 2017a; Tuet et al., 2017b). More
precise measurements of the OP of the different individual SOA are needed in order to evaluate
the overall oxidative potentials of ambient SOA using individual intrinsic OP of different types of
SOA in conjunction with SOA loadings in models. Future studies are needed to develop rational
parameterization schemes for assessing the health risks associated with Nap- and MN-derived
SOA.
**5 Conclusions**
In this study, we investigated the impacts of Nap, 1-MN, and 2-MN oxidation on the
formation of SOC, $O_3$, and radicals from May 20 to June 18, 2018, in the YRD using a revised
CMAQ model and explicit emission inventories. The simulating results of case-1product using the
adjusted emissions (emis-adjust) and a one-product method to fit MN yields best reproduced the
evolution of Nap (NMB=0.01) and MN (NMB=-0.07) when compared with the default case
(NMB=-0.79 for Nap, NMB=-0.85 for MN). The primary sources of Nap and MN were
transportation and residential-related and thus led to a bimodal pattern for their emissions. Whereas
the Nap and MN concentrations were relatively low during the daytime and peaked in the morning,
the generated SOC peaked in the daytime affected by the photochemistry and the evolution of the
boundary layer. All of Nap, 1-MN, and 2-MN had the same trait of contributing much more to
SOC than to SOC precursor emissions, especially for Nap. In general, we found that 3.3% of
aromatic emissions from Nap and derivatives could contribute up to 11.7% SOC generated from
aromatics at the Taizhou site. Nap concentrations reached 27 ppt in the YRD, accounting for 4.1%





and 9.1% (up to 13.7%) of total aromatics emissions and aromatic-derived SOC, respectively. 1-
MN and 2-MN were relatively low, with an average of 3 and 6 ppt in the YRD, and contributed
3.1% of aromatic-derived SOC. At the regional scale, the impacts of Nap and MN oxidation on $O_3$
and radical concentrations were limited. However, substantial increases still occurred in areas with
high Nap and MN emissions and cannot be disregarded. The high SOA formation potential of Nap
and MN and its impact on secondary pollutants proved in this study implied the significance of
such IVOCs except for traditional VOCs when implementing air pollution control policies, energy
use strategies, and health risks evaluation.

**Code and data availability**
The codes used for all the analyses are available on reasonable request to the corresponding author.
All data used in this research are freely available and may be downloaded from the links and cited
references given in the methods section.
**Author contributions**
F.Y., J.L., and J.H. designed the research and conducted the simulations, Y.G., H.W., S.G., and
K.L. collected the observed data. J.A. and C.H. provided emission data. F.Y., J.L., J.H., and M.Q.
analyzed the data, all authors discussed the results. F.Y. prepared the manuscript and all authors
helped improve the manuscript.
**Competing interests**
The authors declare no competing interests.
**Disclaimer**



Publisher's note: Copernicus Publications remains neutral with regard to jurisdictional claims
made in the text, published maps, institutional affiliations, or any other geographical representation
in this paper. While Copernicus Publications makes every effort to include appropriate place
names, the final responsibility lies with the authors.
**Acknowledgements**
This work was financially supported by the National Key R&D Program of China
(2022YFE0136200) and the National Natural Science Foundation of China (No. 42077199).

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



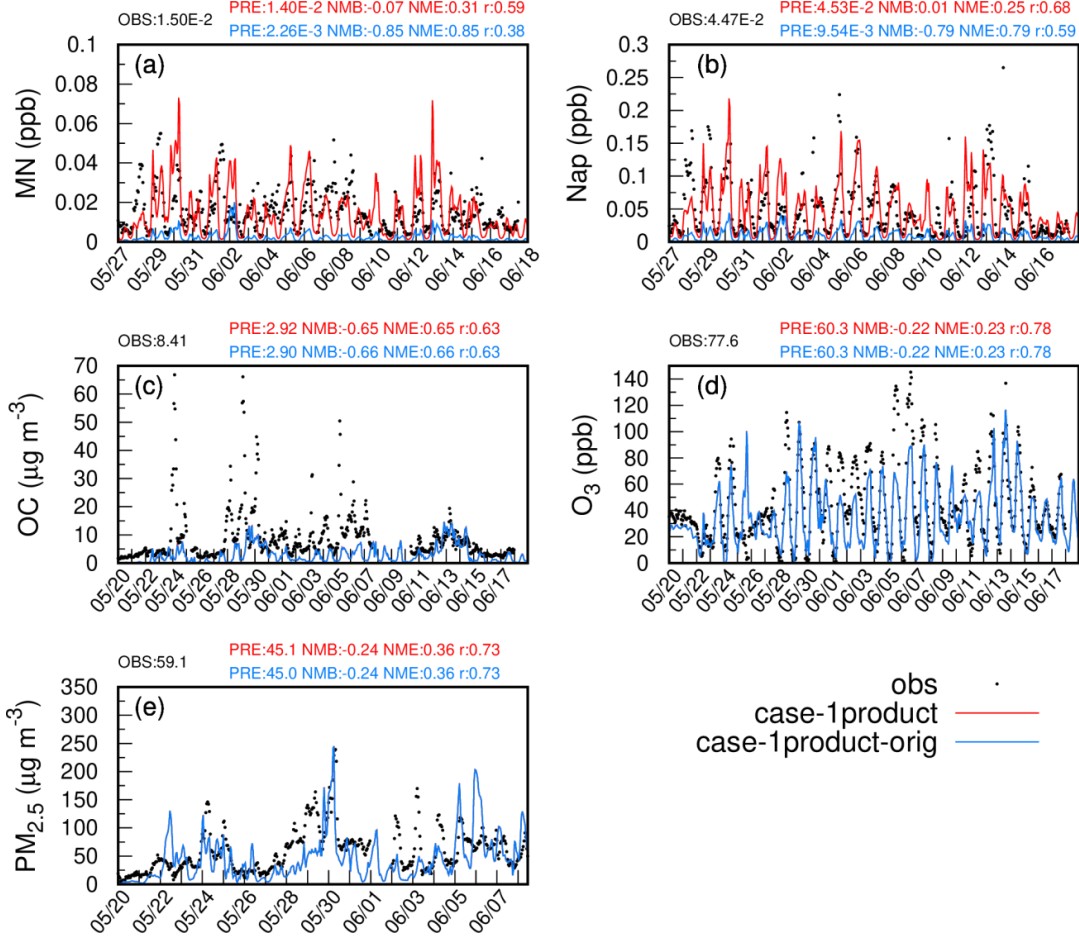

**Figure 1.** Observed and simulated hourly concentrations of MN, Nap, OC, PM$_{2.5}$, and O$_3$ based on emis-adjust (red) and emis-orig (blue) at the Taizhou site. Model performances of daily MN, Nap, OC, PM$_{2.5}$, and MDA8 O$_3$ are shown in blue for case-1product-orig and red for case-1product. OBS and PRE represent averaged concentrations of observations and predictions, respectively.



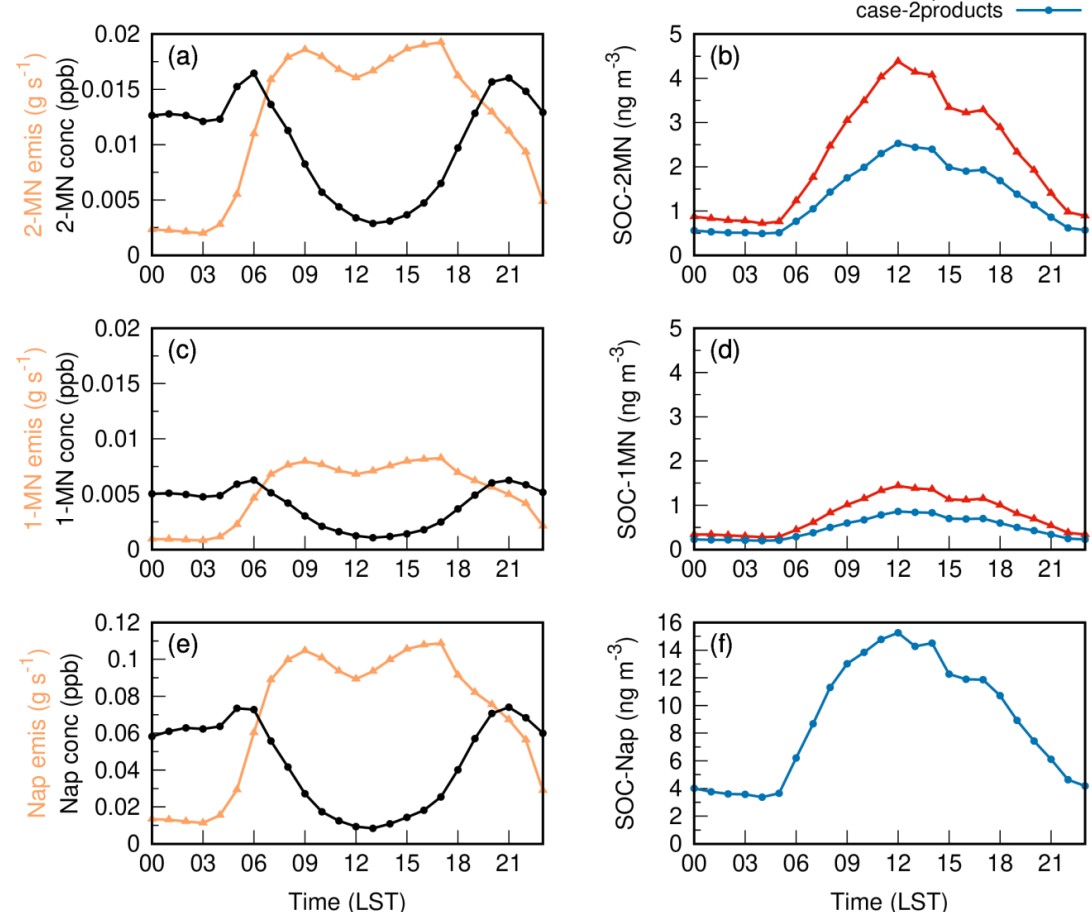

**Figure 2.** Diurnal variations of emissions (yellow line) and predicted concentrations (black line) for 2-MN (a), 1-MN (c), and Nap (e), as well as the corresponding SOC concentrations (b, d, f) at the Taizhou site.



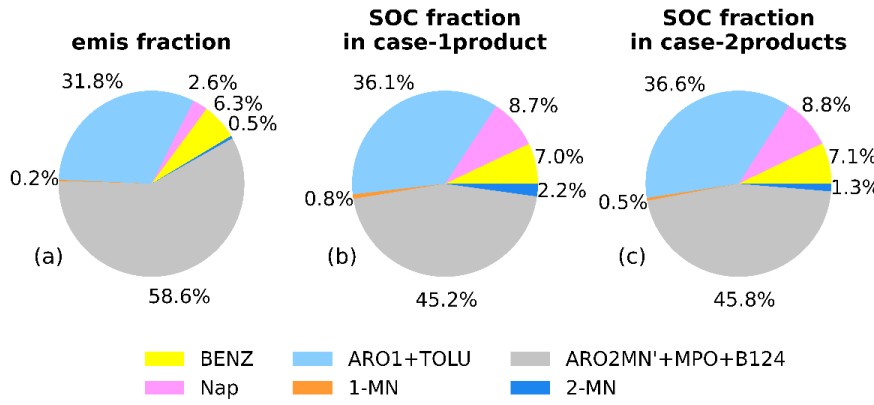

**Figure 3.** Contributions of the major aromatic species to (a) the total emissions of aromatics and the aromatic-derived SOC in (b) case-1product and (c) case-2products at the Taizhou site. These aromatic species are Nap, 1-MN, 2-MN, BENZ, the sum of toluene and aromatics with $k_{OH} < 2 \times 10^4$ ppm$^{-1}$ min$^{-1}$ (ARO1+TOLU), and the sum of xylenes, 1,2,4-trimethyl benzene and aromatics with $k_{OH} > 2 \times 10^4$ ppm$^{-1}$ min$^{-1}$ excluding Nap and MN (ARO2MN'+MPO+B124).



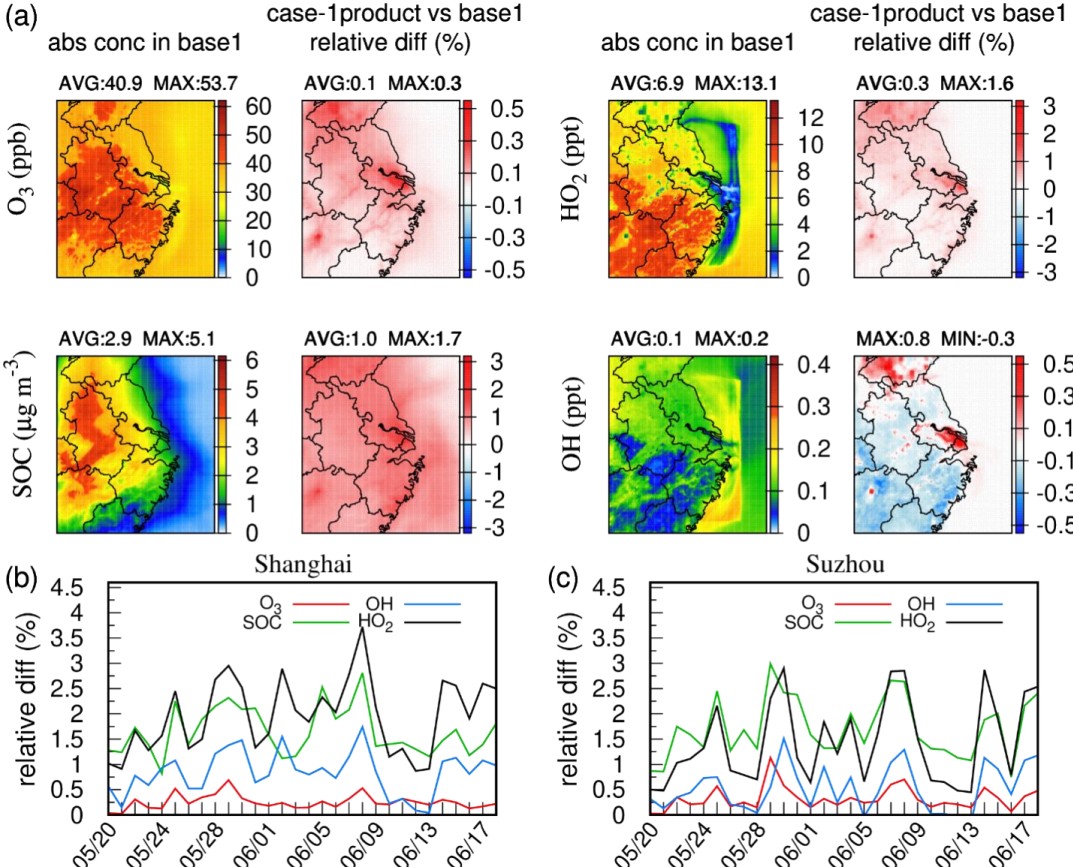

**Figure 4.** (a) Absolute concentrations of SOC, $O_3$, OH, and $HO_2$ in base1, and changes in case-1product relative to base1, respectively. Daily relative changes in case-1product compared to base1 at (b) Shanghai and (c) Suzhou.