# Peer review of "The Role of Naphthalene and Its Derivatives in the"

_EGUsphere, 2023_

## Referee Comment (RC2)

**Comments for "egusphere-2023-3042: The Role of Naphthalene and Its Derivatives in the Formation of Secondary Organic Aerosols in the Yangtze River Delta Region, China" by Ye et al. (2023)**

Ye et al. applied CMAQ model, incorporating revised Naphthalene (Nap) and methylnaphthalene (MN) emissions, as well as including new secondary organic carbon (SOC) formation pathway through 1-MN, to investigate the importance of Nap and MN for SOC formation. They found the model well-reproduced the Nap and MN if using revised emission and one-product SOC formation pathway. Additionally, the findings indicated the oxidation of Nap and MN has negligible effects on regional ozone and radical levels. The draft is well-written, and the topics fit the scope of *ATMOS CHEM PHYS* journal. I enjoy reading the draft, and I only have a few minor comments.

Minor comments:

Line 105: Figure S1 is important for understanding the paper method framework. I checked Figure S1 several times to understand the results. I would recommend moving this figure to the main text. Also, for the SOC formation pathway, the two new added pathways are all about 1-MN. Are there any new pathway for 2-MN in this work? If not, is there any default SOC formation pathway through 2-MN?

Line 166: I recommend moving table S3 to the main text. Additionally, should there be a solid case-1product-orig and base1, and case-1product and case-2product-orig?

Line 193: Why there are no results of case-1product and case-2products in figure 1 (c)-(e) and figure S4 (c)-(e)? Are they overlapped with the original emission simulations? Why do not show the results for the base1 simulation? I think comparison with the base simulation can indicate the importance of the newly added 1-MN SOA formation pathway.

Line 204: Why do you only show the metrics for case-1product in Table S4 and S5? How about the other four cases? If the results from case-1product closely align with measurements, you can simply say that the correlations with observations are higher in case-1product than in the other four cases.

Line 213: Figure 2 is interesting. Did you also check the SOC-Nap diurnal cycle in the base1 simulation? The current results show that the more SOC formation pathways do not indicate more SOC formation. It also depends on the reaction rates of each pathway. So for the default model scheme with only SOC-Nap pathway, which is also the most efficient pathway, may simulate more SOC.

---

## Author Comment (AC1)

**Response to Reviewer #1**

We sincerely appreciate the valuable comments provided by the reviewer, which greatly contributed to enhancing the quality of the paper. Detailed responses are shown below. The reviewer's comments are in regular font, and the author's responses are in red. The changes made to the text are highlighted in blue. The corresponding contents have been updated in the manuscript.

**Reviewer 1:**

Summary:

Naphthalene (Nap) and its derivatives are important intermediate volatile organic compounds (IVOCs) contributing to the formation of secondary organic carbon (SOC). This manuscript uses the CMAQ model to investigate the impacts of Nap and methylnaphthalene (MN) on the formation of SOC and other secondary pollutants in the YRD region. Overall, the manuscript is well-written and easy to follow. The results are interesting and meaningful. I recommend accepting this manuscript after some minor revisions.

Response: Thank you so much for taking the time to thoroughly review our manuscript. Your valuable comments are greatly appreciated and helpful in improving the quality of the manuscript. We have carefully considered your comments and made revisions accordingly. To enhance the clarity, we have changed 'base1' to 'base_zeroNapMN'. Additionally, we conducted a new scenario (base_zeroMN) where the emissions of 1-methylnaphthalene (1-MN) and 2-methylnaphthalene (2-MN) in case-1product were set to zero to quantify the individual impacts of naphthalene (Nap) and methylnaphthalene (MN). We found a mistake in calculating the emissions of Nap and MN from transportation and residential sources in the MEIC inventory for the surrounding area of YRD. Thus, we re-simulated the model using corrected emissions and updated the entire manuscript accordingly. These corrections only have minor influences on the results, and the conclusions remain unchanged. Detailed point-to-point responses are shown below.

Minor comments:

1. Lines 29-30: Does the 3.1% contribution refer to total methylnaphthalene?

Response: Yes, the total contribution from 1-MN and 2-MN was 3.1%. After corrections to the naphthalene and methylnaphthalene emissions from transportation and residential sources in MEIC, the total contribution is 2.4%. We have revised the text as follows:

Lines 29-31: "The concentrations of 1-MN and 2-MN were relatively low, averaging at 2 ppt and 5 ppt. Together, they accounted for only 2.4% of the aromatic-derived SOC."

2. Lines 154-156: Why only the anthropogenic emissions of Nap and MN were scaled

in the emis-adjust case? According to Figure S3, the Nap and MN emissions in the YRD regions are much lower than those in other regions. Could you please show the difference between the MEIC and the YRD emission inventories, and add a brief discussion about such uncertainty?

Response: Thank you for pointing this out. We apologize for the mistake in calculating the Nap and MN emissions from transportation and residential sources in the MEIC inventory, which resulted in an overestimation of the emissions in the surrounding area of YRD. We have corrected this error and updated regional distributions of Nap and MN emissions as shown in Figure R1 (Figure S2 in the revised Supplement). It is worth noting that there are no significant differences between the emissions of YRD and its surrounding regions (MEIC). Since Nap and MN primarily originate from anthropogenic sources (86.7% for Nap and 76.0% for MN), we have only adjusted their anthropogenic emissions in the emis-adjust case.

The model has been re-run with corrected emissions, and the results as well as figures and tables have been updated. Additionally, the text has been revised and a brief discussion about the uncertainty has been added as follows:

Lines 25-27: "Constrained by the observations, anthropogenic emissions of Nap and MN in the entire region were multiplied by 5 and 7, respectively, to better capture the evolution of pollutants."

Lines 158-160: "Considering their predominantly anthropogenic origin, their anthropogenic emissions in the entire region from emis-orig were multiplied by 5 and 7 respectively in the emis-adjust case."

Lines 168-170: "It should be noted that uncertainties associated with the emission inventory and source profiles, which are based on sector-specific mass ratios presented in Table S2, may potentially affect both the distribution and source contributions of Nap and MN."

[Figure]

Figure R1. Regional distributions of Nap, 1-MN, and 2-MN emissions in emis-orig and emis-adjust. SUM represents the total emission rate (tons day$^{-1}$) over the YRD region. MAX represents the maximum emission rate (kg day$^{-1}$) in the grids of the YRD.

3. Lines 160-161: Is the total Nap and MN emission rate over the YRD region 0.9 kg day-1? Please verify the numbers.

Response: Thank you for pointing this out. 0.9 kg day$^{-1}$ refers to the average emission rate of each grid over the YRD region. To enhance clarity, we have calculated the total emission rate of the YRD region and denoted it as 'SUM' in Figure R1 (Figure S2 in the revised Supplement) as above. The text has also been modified as follows:

Lines 162-165: "After adjustments, the total emission rate of Nap and MN in the YRD region in emis-adjust (85.0 tons day$^{-1}$) was approximately 4 times higher than that in emis-orig (18.2 tons day$^{-1}$). The total MN emission rate in the YRD region in emis-adjust was 20.3 tons day$^{-1}$, lower than that of Nap."

4. Lines 195-196: Could you clarify the meaning of "the original settings" mentioned here?

Response: The term "the original settings" refers to the results of case-1product-orig and case-2product-orig simulated with emis-orig that Nap emissions in the YRD were based on the 2017 YRD inventory, while Nap emissions in the rest of the domain and MN emissions of the entire domain were calculated with sector-specific mass ratios and total emissions of non-methane volatile organic compounds. To make this clear, we

have revised the text and Table 1 (Table R1) as follows:

Lines 205-207: "The concentrations of Nap in case-1product-orig and case-2products-orig were significantly underestimated by 79% compared to the observations."

Table R1 Settings of the scenarios.

| Case | Emission setting | SOA parameterization for MN |
|---|---|---|
| case-1product-orig | Nap emissions in the YRD were based on the 2017 YRD inventory; Nap emissions in the rest of the domain and MN emissions in the entire domain were calculated using sector-specific mass ratios and total emissions of non-methane volatile organic compounds (emis-orig) | one-product method |
| case-2products-orig | | two-product method |
| case-1product | The anthropogenic emissions of Nap and MN in the entire domain from emis-orig were multiplied by 5 and 7, respectively (emis-adjust) | one-product method |
| case-2products | | two-product method |
| base_zeroNapMN | Emissions of Nap and MN were set to zero based on emis-adjust | one-product method |
| base_zeroMN | Emissions of MN were set to zero based on emis-adjust | one-product method |

5. Lines 199-200: It is recommended to change the units for 1.40E-2 ppb and 1.50E-2 ppb to ppt.

Response: Thank you for the advice. We have revised the text and replotted Figure 2 (Figure R2) and Figure S3 (Figure R3) based on the updated results as follows:

Lines 209-211: "The modeled concentration of MN by emis-adjust (14.0 ppt) was also comparable to the observed value (15.0 ppt) and showed a good correlation between the two (r=0.59)."

[Figure]

**Figure R2.** Observed and simulated hourly concentrations of MN, Nap, OC, PM₂.₅, and O₃ based on emis-adjust (red) and emis-orig (blue) at the Taizhou site. Model performances for daily MN, Nap, OC, PM₂.₅, and MDA8 O₃ are shown in blue for case-1product-orig and in red for case-1product. OBS and PRE represent the average of observations and predictions, respectively. Note that the red and blue lines overlap in (c)-(e).

[Figure]

**Figure R3.** Observed and simulated hourly concentrations of MN, Nap, OC, PM$_{2.5}$, and O$_3$ based on emis-adjust (red) and emis-orig (blue) at the Taizhou site. Model performances for daily MN, Nap, OC, PM$_{2.5}$, and MDA8 O$_3$ are shown in blue for case-2products-orig and in red for case-2products. OBS and PRE represent the average of observations and predictions, respectively. Note that the red and blue lines overlap in (c)-(e).

6. Lines 201-202: According to Figure 1, the simulated concentrations of OC and PM$_{2.5}$ were nearly identical for both cases. Therefore, the term "improved" may not be appropriate here.

Response: Thank you for your advice. We have modified the text as follows:

Lines 211-213: "For other species, the concentrations of OC and PM$_{2.5}$ were slightly increased in emis-adjust compared to that of emis-orig, although they were underestimated in both scenarios."

7. Lines 269-272: There is no significant difference in the simulated OC concentrations between case-1product and case-2products.

Response: Thank you for pointing this out. We have modified the corresponding descriptions in the main text as follows:

Lines 284-287: "In general, the concentrations of SOC produced by the three PAHs in case-1product were higher than that in case-2products, exhibiting similar spatial distribution patterns in both cases. We will focus on the results from case-1product in the subsequent analysis."

8. Lines 294-302: I'm curious about the diurnal variations in $O_3$ and radicals. Could you provide more details?

Response: Thank you for the advice. The diurnal variations in $O_3$, radicals, and SOC at the two sites have been included in the Supplement as Figure S12 (Figure R4). A brief description of the diurnal changes has been added in the revised text as follows:

Lines 315-318: "It was found that both OH· and $HO_2$· displayed bimodal variations at the two sites, with the most pronounced changes of 0.7–1.0% and 1.6–2.2% occurring in the morning, respectively (Fig. S12). The concentrations of SOC and $O_3$ were elevated in the daytime, reaching peak increments of 2.1–2.3% and 0.4–0.5% at noon."

[Figure]

**Figure R4.** Diurnal relative changes in case-1product compared to base_zeroNapMN in (a) Shanghai and (b) Suzhou.

9. Figure 4: Please check the line length in the color bar ticks.

Response: Thank you for the reminder. We have updated Figure 4 (Figure R5) accordingly, which is now presented as Figure 5 in the revised manuscript.

[Figure]

**Figure R5.** (a) Average concentrations of SOC, $O_3$, OH·, and $HO_2$· in base_zeroNapMN and changes in case-1product relative to base_zeroNapMN. Daily relative changes in case-1product compared to base_zeroNapMN in (b) Shanghai and (c) Suzhou.

---

## Author Comment (AC2)

**Response to Reviewer #2**

We sincerely appreciate the valuable comments provided by the reviewer, which greatly contributed to enhancing the quality of the paper. Detailed responses are shown below. The reviewer's comments are in regular font, and the author's responses are in red. The changes made to the text are highlighted in blue. The corresponding contents have been updated in the manuscript.

**Reviewer 2:**

Ye et al. applied CMAQ model, incorporating revised Naphthalene (Nap) and methylnaphthalene (MN) emissions, as well as including new secondary organic carbon (SOC) formation pathway through 1-MN, to investigate the importance of Nap and MN for SOC formation. They found the model well-reproduced the Nap and MN if using revised emission and one-product SOC formation pathway. Additionally, the findings indicated the oxidation of Nap and MN has negligible effects on regional ozone and radical levels. The draft is well-written, and the topics fit the scope of ATMOS CHEM PHYS journal. I enjoy reading the draft, and I only have a few minor comments.

Response: Thank you very much for taking the time to thoroughly review our manuscript. We truly appreciate your valuable comments, which have been instrumental in improving the quality of our work. Several changes have been made to enhance the clarity of the manuscript. We renamed 'base1' as 'base_zeroNapMN'. Additionally, we conducted a new scenario (base_zeroMN) where the emissions of 1-methylnaphthalene (1-MN) and 2-methylnaphthalene (2-MN) in case-1product were set to zero to quantify the individual impacts of naphthalene (Nap) and methylnaphthalene (MN). The manuscript has been updated with results based on corrected Nap and MN emissions from transportation and residential sources in the MEIC inventory for the surrounding area of YRD. The conclusions remain unchanged. Our study emphasizes the importance of Nap and MN in contributing to secondary organic aerosol (SOA) formation, even with only a small amount emitted into the atmosphere. Although their regional impacts on SOC, ozone, and $HO_x$ radicals may not be substantial, it is crucial to consider their impacts in areas with high emissions such as Shanghai and southern Jiangsu. We have carefully considered all your comments and made revisions accordingly. Detailed point-to-point responses are shown below.

Minor comments:

1. Line 105: Figure S1 is important for understanding the paper method framework. I checked Figure S1 several times to understand the results. I would recommend moving this figure to the main text. Also, for the SOC formation pathway, the two new added pathways are all about 1-MN. Are there any new pathway for 2-MN in this work? If not, is there any default SOC formation pathway through 2-MN?

Response: Thank you for the advice. Figure S1 has been moved to the main text and labeled as Figure 1 (Figure R1). The SOC formation pathways for 1-MN and 2-MN are similar and both are newly added to the CMAQ model. We have revised the figure to

include the SOC formation pathways of 2-MN. Additionally, the reaction of Nap with chlorine radicals has been excluded from the SOC-Nap scheme since it is not considered in the current CMAQ model. The updated figure is shown below:

[Figure]

Figure R1. SOA schemes for naphthalene (Nap), 1-methylnaphthalene (1-MN), and 2-methylnaphthalene (2-MN) in the updated CMAQ model. (a) pre-existing Nap-derived SOA formation pathways fitted by two products under high $NO_x$; (b) newly added SOA formation pathways for 1-MN and 2-MN fitted by two products under high $NO_x$; (c) newly added SOA formation pathways for 1-MN and 2-MN fitted by one product under high $NO_x$. SOA formation from Nap and MN oxidation by OH radicals under low-$NO_x$ conditions is represented by a fixed yield. Parameters for 2-MN are indicated in brackets in (b) and (c). The values of α refer to Table S1.

2. Line 166: I recommend moving table S3 to the main text. Additionally, should there be a solid case-1product-orig and base1, and case-1product and case-2product-orig?

Response: Thanks for the advice. Table S3 (Table R1) has been moved to the main text and labeled as Table 1. Generally, the scenarios can be divided into three categories based on Nap and MN emissions. The first type (emis-orig), i.e. case-1product-orig and case-2products-orig, used default Nap emissions from the YRD inventory and estimated emissions for Nap in the rest of the domain and MN in the whole domain based on source profiles from the SPECIATE database. To improve modeled Nap and MN concentrations, their anthropogenic emissions were adjusted (emis-adjust) according to observed/predicted concentration ratios from aforementioned cases,

named case-1product and case-2products, respectively. The difference between the two cases within each category is the representation of SOA formation from 1-MN and 2-MN, i.e., one-product (1product) or two-product (2products) scheme as shown in Figure R1(b-c). For the third type, i.e., base1, only the emissions of Nap and MN in the entire domain were set to zero on top of emis-adjust. To make this clear, we have renamed 'base1' as 'base_zeroNapMN'. A new scenario (base_zeroMN) was conducted to quantify the individual impacts of Nap and MN by setting the emissions of 1-MN and 2-MN to zero. The revised Table 1 and the text are shown as follows:

Lines 171-184: "Table 1 lists the scenarios conducted in this study. In case-1product-orig, the anthropogenic emissions were based on emis-orig, along with the SOA parameterization for MN fitted by a one-product method in Fig. 1c and that of Nap fitted by a two-product method in Fig. 1a under high-$NO_x$ conditions. To assess the impacts of different SOA parameterizations, case-2products-orig adopted the same setting as case-1product-orig except for utilizing a two-product method for MN-derived SOA under high-$NO_x$ conditions (Fig. 1b). For accurate representations of the fate of Nap and MN in the atmosphere, both case-1product and case-2products employed adjusted emissions (emis-adjust) along with different SOA parameterizations for MN. SOA formation from Nap and MN under low-$NO_x$ conditions in the above cases were all characterized by a fixed yield as shown in Table S1. Overall, the contributions of Nap, 1-MN, and 2-MN to the aromatic SOC were estimated based on different emission inventories and SOA schemes. To evaluate the effects of Nap, 1-MN, and 2-MN on $O_3$, SOC, and radical concentrations, their emissions in case-1product were set to zero and named base_zeroNapMN. A case named base_zeroMN was conducted to quantify the individual effects of Nap and MN by setting the emissions of 1-MN and 2-MN to zero."

**Table R1.** Settings of the scenarios.

| Case | Emission setting | SOA parameterization for MN |
|---|---|---|
| case-1product-orig | Nap emissions in the YRD were based on the 2017 YRD inventory; Nap emissions in the rest of the domain and MN emissions in the entire domain were calculated using sector-specific mass ratios and total emissions of non-methane volatile organic compounds (emis-orig) | one-product method |
| case-2products-orig | | two-product method |
| case-1product | The anthropogenic emissions of Nap and MN in the entire domain from emis-orig were multiplied by 5 and 7, respectively (emis-adjust) | one-product method |
| case-2products | | two-product method |
| base_zeroNapMN | Emissions of Nap and MN were set to zero based on emis-adjust | one-product method |
| base_zeroMN | Emissions of MN were set to zero based on emis-adjust | one-product method |

3. Line 193: Why there are no results of case-1product and case-2products in figure 1 (c)-(e) and figure S4 (c)-(e)? Are they overlapped with the original emission simulations? Why do not show the results for the base1 simulation? I think comparisons with the base simulation can indicate the importance of the newly added 1-MN SOA formation pathway.

Response: Thanks for your insightful suggestion. Yes, the results of case-1product and case-2products (both in red lines) based on adjusted emissions of Nap and MN (emis-adjust) overlapped with those using emis-orig emissions (blue lines), due to minimal impacts on OC, $O_3$, and $PM_{2.5}$. We have added the explanation "Note that the red and blue lines overlap in (c)-(e)" in relevant figure captions (Figures R2 and R3).

We did not separately examine the impacts of Nap and MN on the formation of secondary pollutants since the focus is on the combined impacts of Nap and derivatives. Therefore, the emissions of Nap and MN were set to zero in base1. To clarify this point, we have renamed 'base1' as 'base_zeroNapMN' in the revised manuscript. Additionally, we conducted a sensitivity simulation by excluding 1-MN and 2-MN emissions in case-1product ('base_zeroMN'). The differences between case-1product and base_zeroMN represent the impacts of MN alone, as shown in Figure R4 (Figure S11 in the revised Supplement). Overall, the impacts of Nap on SOC, $O_3$, and radicals are significantly higher than those of MN. The importance of MN is summarized in the revised text as follows:

Lines 294-295: "The impact on $O_3$ was relatively limited, with a maximum increase of 0.3%, primarily attributed to Nap rather than MN (Fig. S11)."

Lines 307-308: "Similar to $O_3$, variations in $OH\cdot$ and $HO_2\cdot$ were primarily influenced by Nap rather than MN (Fig. S11)."

[Figure]

**Figure R2.** Observed and simulated hourly concentrations of MN, Nap, OC, PM$_{2.5}$, and O$_3$ based on emis-adjust (red) and emis-orig (blue) at the Taizhou site. Model performances for daily MN, Nap, OC, PM$_{2.5}$, and MDA8 O$_3$ are shown in blue for case-1product-orig and in red for case-1product. OBS and PRE represent the average of observations and predictions, respectively. Note that the red and blue lines overlap in (c)-(e).

[Figure]

**Figure R3.** Observed and simulated hourly concentrations of MN, Nap, OC, PM$_{2.5}$, and O$_3$ based on emis-adjust (red) and emis-orig (blue) at the Taizhou site. Model performances for daily MN, Nap, OC, PM$_{2.5}$, and MDA8 O$_3$ are shown in blue for case-2products-orig and in red for case-2products. OBS and PRE represent the average of observations and predictions, respectively. Note that the red and blue lines overlap in (c)-(e).

[Figure]

**Figure R4.** Average concentrations of SOC, $O_3$, OH·, and $HO_2$· in base_zeroMN and changes in case-1product relative to base_zeroMN.

4. Line 204: Why do you only show the metrics for case-1product in Tables S4 and S5? How about the other four cases? If the results from case-1product closely align with measurements, you can simply say that the correlations with observations are higher in case-1product than in the other four cases.

Response: Thanks for pointing this out. We evaluated scenarios implementing different Nap and MN emissions, i.e., emis-orig and emis-adjust, against observations. Among them, the results of cases using the same emissions but different SOA schemes for MN were very similar. Considering scenarios using different emissions, the metrics for the case exhibiting the best performances (case-1product) were presented, although the differences between case-1product and case-1product-orig were insignificant. We have included discussions in the text as follows:

Lines 217-218: "It should be noted that the influences of different SOA schemes for MN on the aforementioned species are negligible."

Lines 220-222: "The results of case-1product and case-2products using emis-adjust as the emission data were superior compared to the cases using emis-orig. These findings will be further discussed in the subsequent analysis."

5. Line 213: Figure 2 is interesting. Did you also check the SOC-Nap diurnal cycle in the base1 simulation? The current results show that more SOC formation pathways do not indicate more SOC formation. It also depends on the reaction rates of each pathway. So the default model scheme with only SOC-Nap pathway, which is also the most efficient pathway, may simulate more SOC.

Response: Thank you for your valuable advice. In base1, the emissions of Nap, 1-MN, and 2-MN were all set to zero so that there was no SOC formation from these PAHs.

To enhance clarity, we have renamed 'base1' as 'base_zeroNapMN'. The differences in SOC-MN between case-1product and case-2products are mainly attributed to the yields and equilibrium partitioning coefficients of SVOC products under high-NO$_x$ conditions. When comparing the SOA yields under high-NO$_x$ conditions, 2-MN and 1-MN employing one-product SOA schemes exhibit higher values than Nap (Figure R5). Additionally, the reaction rate of PAHs with OH radicals and their emission rates also affect SOC formation. Overall, among these compounds, Nap demonstrates the highest contribution to SOC. The text has been revised as follows:

Lines 240-247: "Apart from having the highest emissions, Nap also exhibits greater reactivity with OH·. Although its SOA yield under high-NO$_x$ conditions is lower than that of MN fitted by the one-product scheme (Fig. S6), its SOA yield under low-NO$_x$ conditions is the highest among the three PAHs (Table S1). Overall, Nap contributed the most to SOC. 2-MN demonstrates higher SOA yields than 1-MN under high-NO$_x$ conditions in both cases, but a lower SOA yield under low-NO$_x$ conditions. Considering the impact of a higher emission rate (Fig. 3a and 3c), 2-MN contributed two times more SOC compared to 1-MN."

[Figure]

**Figure R5.** Comparison of fitted SOA yield curves of Nap, 1-MN, and 2-MN under high-NO$_x$ conditions with different total organic mass concentrations ($\Delta M_o$). SOA yield ($Y$) is calculated as $Y=\Delta M_o \sum_i \frac{\alpha_i K_{om,i}}{1+\Delta M_o K_{om,i}}$, where values of $\alpha$ and $K_{om}$ come from Table S1.